# Doctor’s Orders—Why Radiologists Should Consider Adjusting Commercial Machine Learning Applications in Chest Radiography to Fit Their Specific Needs

**DOI:** 10.3390/healthcare12070706

**Published:** 2024-03-23

**Authors:** Frank Philipp Schweikhard, Anika Kosanke, Sandra Lange, Marie-Luise Kromrey, Fiona Mankertz, Julie Gamain, Michael Kirsch, Britta Rosenberg, Norbert Hosten

**Affiliations:** 1Institute for Diagnostic Radiology and Neuroradiology, University Medicine of Greifswald, 17475 Greifswald, Germany; 2Institute for Psychology, University of Greifswald, 17489 Greifswald, Germany

**Keywords:** artifical intelligence, machine learning, deep learning, radiology, chest radiograph, chest imaging, X-ray, AI bias, big data

## Abstract

This retrospective study evaluated a commercial deep learning (DL) software for chest radiographs and explored its performance in different scenarios. A total of 477 patients (284 male, 193 female, mean age 61.4 (44.7–78.1) years) were included. For the reference standard, two radiologists performed independent readings on seven diseases, thus reporting 226 findings in 167 patients. An autonomous DL reading was performed separately and evaluated against the gold standard regarding accuracy, sensitivity and specificity using ROC analysis. The overall average AUC was 0.84 (95%-CI 0.76–0.92) with an optimized DL sensitivity of 85% and specificity of 75.4%. The best results were seen in pleural effusion with an AUC of 0.92 (0.885–0.955) and sensitivity and specificity of each 86.4%. The data also showed a significant influence of sex, age, and comorbidity on the level of agreement between gold standard and DL reading. About 40% of cases could be ruled out correctly when screening for only one specific disease with a sensitivity above 95% in the exploratory analysis. For the combined reading of all abnormalities at once, only marginal workload reduction could be achieved due to insufficient specificity. DL applications like this one bear the prospect of autonomous comprehensive reporting on chest radiographs but for now require human supervision. Radiologists need to consider possible bias in certain patient groups, e.g., elderly and women. By adjusting their threshold values, commercial DL applications could already be deployed for a variety of tasks, e.g., ruling out certain conditions in screening scenarios and offering high potential for workload reduction.

## 1. Introduction

Chest radiography has been and remains one of the most commonly used imaging modalities to this day. Due to its minimal necessary equipment and low radiation exposure, thoracic radiography is ubiquitarily and readily available and performed millions of times daily all around the world to provide information on a wide array of thoracic diseases [1,2,3]. The potential impact of high quality thoracic imaging becomes all the more evident when considering the leading causes of death worldwide in the 2019 WHO report. With four conditions of heart and lungs present among the top ten causes, they alone account for approximately 30% of all fatal diseases [4]. Similar numbers have been found for emergency departments in the USA in 2018 by Weiss et al. and in Germany in 2017 by Reins et al., where patients with diseases of the thoracic organs accounted for up to 29.9% of overall admissions and even up to 46.9% in the most urgent triage group [5,6].

Considering the clinical importance of thoracic diseases, chest radiography today is not only used as a tool to confirm clinical diagnoses but recently even more so as a means to rule out potentially life-threatening diseases as part of the sought after zero mistake policy in modern medicine. The improved availability of radiological imaging and sociodemographic developments such as aging populations and the associated increasing morbidity rates, as well as the COVID-19-pandemic, have also led to ever-rising numbers of radiological examinations over the last years, thus pushing radiology departments all around the world to reach or even go beyond their capacities and bearing risk for the burnout of radiologists [7,8,9,10,11,12].

In light of all of these factors and the massive potential for improvement, there has been a lot of attention by researchers and companies in radiology towards developing solutions, especially focusing on engineering, mainly deep learning (DL) (a special type of machine learning (ML))-based artificial intelligence (AI) models for automated image classification [13,14,15,16]. Radiology has been and is still at the forefront of the general trend of the implementation of medical AI applications, which have been on the rise significantly since around 2018. As of today, around 700 medical AI applications have been FDA approved, about 530 of which are designed for tasks in the radiology subspecialities [17].

However, most of these applications are designed to perform very specific tasks, e.g., the detection of nodules, thereby considerably limiting their potential clinical impact [18]. Such an algorithm might perfectly report all cases of one specific disease while simultaneously ignoring all other abnormalities due to a lack of training to identify and report other findings.

Plesner et al. in 2023 presented a promising and slightly different approach for a DL application, thus elegantly avoiding this dilemma [19]. While still performing a very specific task, the DL application in their study classified images into “normal” and “abnormal”, thereby summing up all potentially pathological cases. This bears high potential for workload reduction, especially in ruling out diagnostics, since a standardized report could be automatically created for all “normal” images. However, it leaves “abnormal” scans virtually uninterpreted and therefore still requiring a substantial amount of human supervision.

Addressing this problem, another promising approach includes DL applications that have been trained on a broad spectrum of common and especially life-threatening diseases. In theory, they bear the potential to rule out healthy patients while simultaneously providing comprehensive interpretation for pathological scans. First explorations into this matter have only recently been presented by Yoon et al. in 2023 [20]. However, as of now, most of these applications have barely been tested in the clinical setting, and only around 2.1% of medical AI devices have been tested by third parties [17].

Considering the importance of thoracic imaging, the aim of this study was the clinical evaluation of a commercial tool designed to automatically detect a wide spectrum of essential conditions in chest radiographs and to analyze its performance in different application scenarios, both for diagnosis of one single abnormality and for comprehensive diagnosis of multiple pathological features. This could potentially have a big impact on clinical routine. Automatically excluded examinations of healthy patients would either no longer need to be interpreted by radiologists at all or at least not in a timely manner, thereby substantially reducing radiologists’ workload while simultaneously supporting them in reporting findings in sick patients. Therefore, another goal of this study was to estimate the potential workload reduction for different settings of the AI application.

## 2. Material and Methods

The implementation of this monocentric, retrospective study was approved by the local ethics committee (reference BB 139/19). The requirement for written informed consent was waived. No additional scans were performed for this study, and all included chest radiographs were clinically indicated. The authors are in no way associated with the company and maintained control of all data and additional information of this presented study.

### 2.1. Study Population

A total of 519 chest X-rays in two planes were acquired for the study. The radiographs were consecutively performed at a German university hospital over a four-month period in 2016. The time span between the examination and this study was chosen, as after this period no significant clinical consequences would be expected from potential changes in findings. Only examinations of patients older than 18 years were included, as there are differences in the interpretation of X-rays between children and adults for several pathological findings [21]. Suitable examinations were identified using the modality filter “chest radiograph 2 planes, adult” of the Picture Archiving and Communication System (PACS; Agfa Healthcare, Mortsel, Belgium) and then exported after pseudonymization. The flowchart in Figure 1 illustrates further processing of the images. A total of 42 patients were excluded, as detailed in Table 1. The study cohort for final analysis comprised 477 chest examinations in two planes.

### 2.2. Acquisition of Radiographs

All examinations analyzed in this study were conducted using the same X-ray machine (DigitalDiagnost C90, Philips Healthcare, Amsterdam, The Netherlands) at a tube voltage of 125 kV, a tube-detector distance of 1800 mm, and a detector resolution of 2900 lines and 2456 columns (Pixium 4600, Philips Healthcare, Amsterdam, The Netherlands). Images were acquired in the DICOM format. Alongside the images, the clinical information of referring departments was extracted. These consisted of a brief, keyword-style case overview with essential details to facilitate radiological interpretation and potential inquiries in the process. From the DICOM files, patients’ sex and age were also extracted.

### 2.3. Image Analysis

#### 2.3.1. Gold Standard

While most radiological AI studies retrospectively use the information collected from pre-existing final image reports from clinical routine as their reference standard, for this study, the decision was made to establish the gold standard by reiterating the four-eyes principle in clinical reporting. Therefore, independent from AI image analysis, the radiographs were interpreted by two experienced radiologists (over 5 and 20 years of experience) working autonomously at different radiological workstations.

Radiologists had access to both PA and lateral images, as well as the clinical information from the examination request. Using a checkbox form (Appendix B) for each radiograph, examiners could indicate which of the investigated abnormalities were present. Furthermore, it was noted whether foreign material (such as ECG electrodes, pacemakers, etc.) was depicted in the images. The form also included the option to specify any imaging errors that might justify exclusion from the study. After excluding examinations that could not be evaluated without limitations, a total study population of 477 patients remained.

After initial independent evaluation of all images by the two radiologists, an interim analysis of the results was conducted. All examinations were identified for which discrepancies had arisen between the findings of the two radiologists. Using newly created evaluation forms, these images were subsequently jointly re-evaluated, and a consensus finding for the gold standard was established. The examinations for which concordance had already been established during the initial reading were incorporated into the gold standard without further changes, as depicted in Figure 1.

#### 2.3.2. DL Analysis

After pseudonymization and application of the exclusion criteria, the remaining 477 examinations were transferred in the DICOM format from the PACS of the university hospital to an independent on-site server of Infervision Medical Technology Co., Ltd. (Beijing, China) within the radiology department. On this server, image analysis was performed using the InferRead DR Chest software (version 1.0.0, not CE certified at the time of the study). This is a commercial application for automated interpretation of chest X-rays and is based on a convolutional neural network (CNN), which is a special type of machine learning. For machine learning in this type of data processing, the native algorithm undergoes initial training using a large number of annotated (image) data [22,23].

Processing, named in reference to the structure of biological neural networks, is performed in multiple layers, where for each layer different features (concerning e.g., shape or density of a lesion) of the data are identified, which might statistically be correlated with the specific abnormality [24]. During the training process, nodes (the CNN’s “neurons”) of the individual layers of the algorithm develop specifically linked structures based on statistical weighting, thus being capable of recognizing identical or similar patterns in other images to use for classification. Compared to the “classic” rule-based programming, where developers have to specify each individual step in the detection of lesions within the algorithm, machine learning has the significant advantage in that it enables the computer to autonomously develop criteria by which it can classify pathological changes. The InferRead DR Chest algorithm used in this study is based on the “U-Net”-architecture, which was introduced by Ronneberger et al. in 2015 [25]. The U-Net structure is a special type of convolutional neural network that was specifically designed for biomedical image segmentation. It applies both contracting and expansive layers to adjust image resolution for improved and more efficient data processing compared to other convolutional neural networks. The implementation of skip connections links the convolutional layers within the architecture, thereby facilitating information transfer within the algorithm. The core algorithm of the specific application evaluated in this study consists of five contracting and expansive layers and is described in detail in Table 2. This specific software was trained for the detection of a total of 17 different abnormalities, which are listed in Appendix A. The training dataset for each type of lesion consisted of more than 20,000 annotated images.

Due to their clinical relevance this study focused on evaluating findings for the following seven abnormalities: adenopathy, atelectasis, fracture, pleural effusion, pneumonia, pneumothorax and tumor. In the DL reading, InferRead DR Chest provided a probability for every single abnormality in all of the chest radiographs. Probability values range from 0 (the abnormality is very unlikely present in this image) up to 1 (the abnormality is almost certainly present in this image). It is common for machine learning applications to provide a probability score to back up their findings and give users at least some opportunity of interpretation of the softwares’ otherwise completely blackboxed results. InferRead DR Chest also provided an image file of the examination and a written report. Upon surpassing an adjustable threshold, the identified finding was graphically highlighted in the image file and included into the report, as can be seen in Figure 2.

In the next step, results of the gold standard were matched with the DL results for further evaluation. For positive findings, concordance of DL application and the radiologists in the gold standard is best for higher DL results. Lower DL results indicate better concordance for gold standard negative cases.

### 2.4. Statistical Analysis

For continuous data, mean and standard deviation (SD) or—in the case of non-normally distributed data—median and interquartile range (IQR: Q1–Q3) are provided, while categorical data are presented with absolute and relative frequency. Sensitivities and specificities for the DL findings of each abnormality were evaluated using receiver operator characteristics (ROC), and the results are given in percentages, the area under the curve (AUC), and its 95% confidence interval (95%-CI) are given as well.

Additionally, by calculation of the highest Youden’s index (Youden’s J = sensitivity + specificity − 1), optimized thresholds for categorizing a positive or negative finding using the DL software were determined. Using this information, the amount of scans concordantly labeled was evaluated. The same procedure was performed for cutoff values with a sensitivity of 95% or higher in the exploratory ROC analysis.

For gold standard positive cases, DL results above the specific cutoff thresholds are hereinafter referred to as “true positive”, while those below the thresholds are referred to as “false negative”. Gold standard negative cases with DL results below the cutoff values are referred to as “true negative”, and those above the thresholds are referred to as “false positive”. The positive predictive value (PPV), negative predictive value (NPV), and false positive rate (FPR) are given as percentages.

Additionally, hierarchical regression was performed to investigate the influence of sex, age, comorbidity, and the presence of foreign material on the deviation between the gold standard and the DL findings.

Microsoft Excel (Microsoft Corporation, Redmond, Washington, DC, USA), SPSS (IBM, Armonk, New York, NY, USA), and the matplotlib package for Python by Hunter were used for statistical analysis [26,27,28].

## 3. Results

### 3.1. General Patient Characteristics

After application of exclusion criteria, the final study population contained 477 chest radiographs in two planes. It included 284 men (59.54%) and 193 women (40.46%) between 18 and 91 years old, with a mean age of 61.4 years old (SD: 16.73 years) and a median age of 63 years old (IQR: 51–75); the details are presented in Table 3.

The excluded group consisted of 42 patients. As seen in Table 1, most of the patients were excluded due to incomplete lateral image acquisition. Two of the main reasons for incomplete radiographs tended to be patients’ height and girth, which are statistically larger in male patients, thereby possibly explaining the percentage of men in the exclusion group exceeding their overall portion in the general study population [29].

### 3.2. Gold Standard

During the first individual reading, radiologists achieved concordant results for 274 (57.44%) patients, thus reporting 107 abnormal findings. During the consensus reading, agreement was reached for another 119 findings from the remaining 203 patients. Detailed results of the distribution for findings of each specific abnormality are presented in Table 4 and Figure 3.

For the final gold standard, radiologists reported a total of 226 positive findings in 167 out of 477 patients (35.01%). The most commonly found pathological finding was pleural effusion, which was reported in 66 (13.84%) patients, while the least frequent finding was pneumothorax, which was reported in six (1.26%) cases.

Most radiographs (310 patients, 64.99%) did not contain any of the investigated abnormalities. A total of 116 (24.32%) patients presented one finding, 44 (9.22%) patients presented two findings, 6 (1.26%) patients presented three findings, and only 1 patient (0.21%) presented four findings in the final gold standard. Gold standard positive patients were on average older, with an average of 68.6 years and medians ranging from 61 years (pneumothorax) to 74 years (fracture). Gold standard negative patients showed an average age of 62.1 years and median ages ranging from 61 (pleural effusion) to 63 years (adenopathy). Gold standard positive and negative findings showed similar distributions in both men and women.

Both radiologists also evaluated the presence of external material in the images during their first reading, thus finding material in 191 (40.04%) patients and leaving 286 (59.96%) patients without foreign material depicted.

### 3.3. DL Results

For gold standard positive cases, the DL results did not show normal distribution. On average across all abnormalities, the DL probabilities showed a median of 0.6047 (IQR: 0.4489–0.7601). The highest DL results were seen in the gold standard positive pneumonia cases, with a median probability of 0.6953 (IQR: 0.5705–0.8207). The lowest DL results were found in the gold standard positive atelectasis cases, with a median probability of 0.5065 (IQR: 0.428–0.6619).

The detailed results for the gold standard positive cases are presented in Table 5. Figure 4 presents a graphic representation of the distribution of DL results. It also includes the cutoff values used for classification of the DL values into “positive” and “negative” for both of the cutoff values by applying the optimization of Youden’s index and also the more conservative approach applying sensitivities of 95% or higher in the exploratory analysis, thus resulting in lower cutoff values.

The DL results for the gold standard negative findings did show normal distribution, but for reasons of comparability, the median probabilities for these cases are given in Table 6, as well as the mean value. The DL probabilities across all abnormalities showed an average mean of 0.3099, with a standard deviation of 0.1757. The average median across all abnormalities was 0.2625, with a Q1 of 0.1823 and a Q3 of 0.3962. The lowest DL results for the gold standard negative cases were seen in pleural effusion, with a mean of 0.1581 (SD: 0.1411) and a median of 0.1048 (IQR: 0.0612–0.1940). The highest DL results were found for the gold standard negative cases of adenopathy, with a mean of 0.4668 (SD: 0.1311) and a median of 0.4556 (0.3696–0.5599).

### 3.4. ROC Analysis

#### 3.4.1. Optimization by Youden’s Index—Single Entity Reading

For evaluation of the level of agreement between the DL application and gold standard reading, an ROC analysis was performed for every entity of abnormal findings; the detailed results are presented in Table 7. An average AUC of 0.842 (0.76–0.92) across all entities was found, thereby ranging from 0.791 (0.73–0.86) in tumors up to 0.92 (0.89–0.96) in pleural effusion. ROC curves for fractures and pleural effusion are presented in Figure 5.

Table 7 also provides the values for sensitivity and specificity and states the numbers for correct positive, correct negative, false positive, and false negative classification, as well as the positive predictive value (PPV), negative predictive value (NPV), and false positive rate (FPR) of the DL reports for every single abnormality, which were classified using the threshold values for classification determined by optimization by Youden’s index.

In this application scenario, the sensitivity and specificity averaged at 85.33% and 75.1%, respectively. The highest sensitivity was found in adenopathy with 100%, and the highest specificity was found in pleural effusion with 86.4%. The lowest sensitivity values were found in fracture with 69.7%, and the lowest specificity was found in adenopathy with 58%.

False negative cases rangeed from 0 cases for adenopathy to 10 cases for fractures. On average across all entities, about five cases that would have been marked positive in the gold standard were classified as negative by the DL application, which came out to 1.05% of all cases included in the study population. True negative cases ranged from 248 in tumors to 415 in pneumothorax, with an average of 333.71 cases across all pathological entities, which came out to 69.96% of the study population.

For the positive DL results using the optimized cutoff value, the numbers of true positive findings ranged from 5 for pneumothorax up to 57 for pleural effusion. Across all entities, the number of true positive cases amounted to an average of 27.29, which came out to 5.72% of all study patients. False positive cases ranged from 56 in pleural effusion and pneumothorax to 195 cases in adenopathy, with an average of 111 patients being classified false positive, thereby making up 23.27% of all cases on average.

PPVs ranged from 6.25% in adenopathy to 50.44% in pleural effusion, with an average of 21.05% across all pathological findings. NPVs showed numbers between 97.33% for fractures and 100% for adenopathy, with an average of 98.54%, and FPRs ranged from 11.89% in pneumothorax to 42.86% in tumor, with an average of 24.96%.

#### 3.4.2. Optimization by Youden’s Index—Combined Reading

For the combined reading of all seven examined pathological entities using the cutoff values optimized by Youden’s J, the DL application and gold standard showed concordance for a total of 78 (out of 477, 16.35%) patients.

Out of these patients, 62 (13%) were correctly diagnosed without any findings, and 16 (3.35%) patients with one or more positive diagnoses were diagnosed completely right. There were 364 (76.31%) patients with one or more false positive findings. For 35 patients (7.34%) with false negative DL results, each one positive finding had been missed. The numbers for the combined comprehensive reading are summarized in Table 8 for the optimized cutoff values and exploratory analysis.

The sensitivity for the combined reading of all seven entities was 31.37%, and the specificity was 14.55%. The PPV was 4.21%, with an NPV of 63.92% and an FPR of 85.45%. Figure 6 shows two exemplary cases for patients with false positive and false negative classification. Results for comprehensive reporting of all evaluated pathological entities compared to results when only screening for pleural effusion are presented in the pie charts in Figure 7.

#### 3.4.3. Exploratory Analysis—95% Sensitivity

By applying sensitivity rates of 95% or higher, the cutoff values presented in Table 9 were calculated. When scanning for only one kind of abnormal finding using these cutoff values, on average around 31 (6.5%) of all cases were classfied true positive, 197 (41.3%) true negative, 248 (51.99%) false positive, and 1 (0.21%) case false negative.

The numbers of true positive cases ranged from 6 in pneumothorax to 63 in pleural effusion; true negative cases showed numbers between 142 for pneumothorax and 259 for pleural effusion. For false positive cases, the numbers ranged between 152 for pleural effusion and 355 for fracture, and for false negative cases, they ranged between 0 for adenopathy and pneumothorax and three for pleural effusion.

The PPVs ranged from 1.79% for pneumothorax to 29.30% for pleural effusion, with an average of 11.16% across all pathological entities. The NPVs were between 98.85% for pleural effusion and 100% for adenopathy, with an average of 99.42%, and the FPRs ranged from 42.03% for adenopathy to 79.95% for fractures, with an average of 55.77%.

For the comprehensive reading of all seven abnormal entities simultaneously using cutoff thresholds for 95% sensitivity for each individual pathological entity, there were completely correct reports for eight (1.68%) patients. A total of 3 (0.63%) patients received completely true reports with positive findings, 5 (1.05%) patients were correctly diagnosed without any findings, 461 (96.65%) patients had one or more false positive findings, and in 8 (1.68%) patients each one finding was a false negative. The sensitivity was 27.27%, with a specificity of 1.07%, and the PPV for the comprehensive reading was 0.65%, with an NPV of 38.46% and an FPR of 98.93%.

### 3.5. Patient Characteristics and Their Influence on DL Results

The DICOM files that were used for this study included the basic patient specific information of age and sex. In the gold standard reporting process, additional information on the presence of external material was obtained, and the analysis of entities within the scans provided information on multimorbidity for patients with different findings. To evaluate the influence that those patient-specific characteristics have on DL results, logistic regression was performed separately for both the gold standard negative and positive findings.

Across all entities for gold standard negative cases, a combined predictive effect of $R^{2}$ = 0.042 (*p* < 0.01) was found for the parameters of sex, age, multimorbidity, and foreign bodies. Significant effects were found for the parameters of multimorbidity (β = 0.151, *p* < 0.001), age (β = 0.089, *p* < 0.001), and sex (β = −0.038, *p* = 0.031; sex was dummy coded using “0” for male and “1” for female).

For the gold standard positive cases across all abnormal entities, the combined predictive effect was $R^{2}$ = 0.018 for the regression model, but the model’s finding was not significant (*p* = 0.093). A significant effect was found for the parameter sex (β = 0.153, *p* = 0.026). There were no significant effects for the parameters multimorbidity and age in the gold standard positive findings and no significant effect for the presence of external material in the images for neither the gold standard positive nor negative cases.

## 4. Discussion

For evaluation against two radiologists as a reference standard, the DL- application analyzed in the presented study achieved a very good average AUC of 0.842 (0.76–0.92) across all entities when screening for one single disease. The AUCs in this scenario ranged from good results of 0.791 (0.76–0.92) for tumors to excellent results of up to 0.92 (0.89–0.96) for pleural effusion. The results of this study align with the findings from comparable examinations of other DL applications for automated detection of pathological findings. The sensitivities and specificities, by applying the threshold values optimized by Youden’s index, showed an average of 85.33% and 75.1% across all entities, respectively, which can be considered good and align with the results from most recent studies screening for one specific disease as well. The AUC of the ROC, sensitivities, and specificities in those studies ranged from 0.89 to 0.97, 63 to 90%, and 96 to 100% for pneumothorax, respectively, 0.94 to 0.97, 62 to 95%, and 65 to 100% for pleural effusion, respectively, were 0.92, 93.2%, and 79.4% for rib fractures, respectively, ranged from 0.66 to 0.72, 46 to 67%, and 77 to 86% for (COVID-19) pneumonia, respectively, and reported an ROC of 0.81 for lung nodules [30,31,32,33,34].

The goal of optimization by Youden’s index, which was applied for cutoff calculation in this study, was to balance the influence of the sensitivity and specificity on the cutoff value. This worked well for pleural effusion, where the sensitivity and specificity each amounted to 86.4% for the optimized threshold value. There are, however, limits to this method of optimization, if either the sensitivity or the specificity outweighs the other one strongly. This could be seen in the entities fracture and tumor, thereby showing the two sides of the method’s limitations: For fractures, the specificity (82%) was remarkably higher than the sensitivity (69.7%) and therefore more dominant in the optimization using Youden’s index. While relatively few false positive findings were being reported, the resulting higher cutoff value led to gold standard positive findings possibly being overlooked (creating false negative reports). For potential tumors, the sensitivity (88.4%) outweighed the specificity (57.1%), thereby resulting in a lower cutoff value. For most screening situations, this approach might be more applicable, because the aim often is to reduce false negative findings, which bear the highest risk of potential harm to patients.

When evaluating only results of one single entity for cutoff values optimized using Youden’s index, true negative findings on average made up about 333.7 (75.1% of all gold standard negatives and 69.97% of the total) cases, with only 5 (about 1.46% of negative findings and 1.05% of all cases) false negative findings. The PPVs were relatively low, but results showed excellent reliability of negative reports with remarkably high NPVs of at least 97.33% (fracture) and an average of 98.54%. These numbers show very high potential for significant workload reduction in automated ruling out of one specific disease.

In the exploratory analysis, we aimed for sensitivities above 95% for every single pathological entity. Even for this conservative application scenario, true negative reports still made up about 44.46% of the gold standard negative cases (41.24% of all cases) on average. The NPV showed rates of at least 98.85% (pleural effusion) and an average of 99.42%. Depending on the reproducibility of these results, those remarkably high NPVs might enable radiologists to significantly reduce their workload even when screening for a potentially life-threatening disease while maintaining patient safety. The potential for workload reduction, however, dropped significantly when combining the results from each abnormal finding to a simultaneous comprehensive reading of all seven evaluated entities due to accumulation of error values of the individual readings. For the combined readings using cutoff values optimized by Youden’s J, only 78 patients (out of 477, 16.35%) were diagnosed completely right. A total of 62 (13%) of these patients were correctly diagnosed without any findings, with only 16 (3.35%) patients with one or more positive diagnoses being diagnosed completely right. While there were 364 (76.31%) patients with one or more false positive findings, in 35 patients (7.34%), each one positive finding had been missed. The findings for these cutoff values would allow for some workload reduction of about 13% but at the cost of missing findings in around 7.34% of patients, which is not acceptable, as the minimum goal of applying DL tools in the clinical setting should be noninferiority to the current gold standard to avert potential danger for patients. Therefore, for the evaluation of classification in this study, a conservative approach was applied: False negative cases included all patients with one or more true positive findings who had additional findings missed by DL classification. This applied to 13 of the 35 false negative cases. Similarly, the aforementioned applied to false positive cases, which included every patient with no missed finding but one or more false positive findings regardless of possible additional true positive findings. As patients with positive reports should always be double-checked by radiologists, those cases would have needed human supervision and could not have been removed from radiologists’ worklists either way. In the exploratory analysis at sensitivity rates of 95% or higher, false negative numbers in the combined reporting could be reduced to eight (1.68%). However, in doing so, the number of true negative cases dropped to five (1.05%), thereby deteriorating the ratio of false negative to true negative cases and leaving virtually no remaining workload reduction when the software was used unsupervised for ruling out healthy patients at this sensitivity level.

This study’s results for potential workload reduction by ruling out diseases were lower than those presented by Yoon et al. in 2023, who used a different approach on the automated triage of normal chest radiographs in a total of 1964 outpatients, which is a patient population with considerably lower morbidity than inpatients [20]. They employed a pattern of DL applications that screened for nine different types of abnormal findings (atelectasis, consolidation (pneumonia), nodules (tumors), pleural effusion, pneumothorax, as well as calcification, cardiomegaly, fibrosis, and pneuperitoneum). Cases were classified into “normal”, “clinically insignificant”, and “clinically significant” by their reference standard. The DL application achieved 5% noninferiority for a workload reduction of up to 70%, thus sorting patients regarding their DL scores and only having radiologists report the 30% of examinations with the highest scores. However, the authors applied considerably lower sensitivities, stating that a sensitivity of 80% would be clinically acceptable and therefore using significantly less strict cutoff values than in the presented study. The prevalence of sick patients was also considerably lower, with 12% as opposed to 35% in the presented study. Similar results as in this presented study were found by Dyer et al. in 2021. For an inpatient study collective with a comparable prevalence rate of about 41% patients with gold standard positive findings, they were able to automatically remove about 15% of the patients from the worklist [35]. Worklist reduction could be improved to 18.5% in a follow-up study by the same group in 2022 [36]. The authors in those studies also used a different approach of automated triage, with a DL application classifying images without findings as “high confidence normal” without specifying any pathological findings in the “abnormal” subgroup. This approach was applied by Plesner et al. as well in 2023, who evaluated a commercially available DL algorithm that was able to autonomously report on 28% of normal posterioranterior chest radiographs with a sensitivity above 99% for any abnormality, thereby reducing overall workload by 7.8% [19].

The impact of AI workload reduction might be improved when combining different applications into a layered system. Algorithms as presented by Dyer et al. and Plesner et al. that simply classify images into “normal” and “abnormal” without any specific diagnosis might be a suitable first layer for automated triage. In a next step, DL software such as presented in this study could be applied as a second layer, thereby generating specific diagnoses for the “abnormal” cases using tailor-made cutoff values, which at best can be validated by test data for every institution. Even more so, as there were remarkable differences for cutoff values depending on the pathological entity and preferred method of optimization, as presented in this study. In consultation with referring physicians, modern radiologists should evaluate the required concept of DL aided diagnostics regarding clinical questions. Thresholds for the classification of DL results into positive and negative findings then need to be adapted for each entity and desired application purpose. When using DL applications for the confirmation of highly likely clinical diagnoses, relatively high cutoff values could be applied to avoid large numbers of useless false positive findings. When screening for specific diseases in large patient populations on the other hand, lower threshold values should be applied to minimize the risk of false negative findings, thereby potentially putting patients at risk. Lower classification thresholds, however, will lead to increasing rates of false positive findings. While this is mostly tolerated for the sake of improved sensitivity, false positive results can cause anxiety in patients and possibly lead to unnecessary or even harmful treatment of the falsely diagnosed diseases [37]. Regarding these insecurities, radiologists and policy makers should develop standards of communication for the utilization of AI and interpretation of automated DL results to ensure that the application of AI in the clinical routine complies with good medical practice and minimizes harm to patients.

In this study, we also evaluated factors possibly affecting the concordance of radiologists’ findings in gold standard and DL results. Like the algorithm presented by Schultheiss et al., the DL application evaluated in the presented study showed no significant effect on the results for the presence of external material on the radiographs [38]. This is fortunate, since the percentage of images containing foreign bodies was remarkably high at about 40% and is much likely to further increase when considering aging populations with a larger demand for (implanted) medical support devices [39]. The regression model, however, did show moderate effects of the factors of multimorbidity, age, and sex on the magnitude of the DL findings. Patients with more abnormal findings, a higher age, and male sex were more likely to receive higher DL results for (additional) pathological entities, for which gold standard classification was negative. Therefore, these patient subgroups appear to be more prone to “false positive” findings. For gold standard positive cases, the regression model showed a moderate effect of sex in the other direction. In this case, female patients were more likely to receive lower DL results, thus increasing the risk for false negative classification. Even though the regression model for gold standard positive cases was not statistically significant due to smaller numbers of positive than negative cases, it still indicates the tendency to discriminate against female patients, thereby potentially putting them at risk for diseases being missed. The findings from this study differ from Ahluwalia et al., who found worse concordance between radiologists and an AI application in patients with only one solitary finding and with younger age, specifically in DL-aided reporting of chest radiography [40]. The presented results of bias in classification based on sex, however, do coincide with the majority of studies on bias in medical AI applications, thereby indicating discrimination against marginalized and underrepresented subgroups such as female patients and people of color [41,42]. Due to its systemic implications, bias in AI development and research is an issue that needs to be addressed imperatively [43,44,45,46,47]. Especially in training datasets, the lack of representation of patient groups may lead to or catalyze institutionalized discrimination [48]. The study population of this group, though continuously acquired, by chance also contained significantly more male than female patients, which limits its explanatory power for female patients.

To the best of the authors’ knowledge, this study is the first one that analyzed the performance of a commercial DL application for automated comprehensive interpretation of a wide spectrum of abnormal findings in chest radiographs in a middle-European patient population. To their knowledge, it is also the first clinical study evaluating possible fields of application scenarios of the same DL software depending on applied cutoff thresholds.

This study covered seven types of abnormal findings representing an extensive spectrum of diseases with high clinical relevance. Due to this approach, no patients in the examined image modality had to be excluded for medical reasons but only for examination errors. However, the focus on chest imaging in two planes naturally excluded all bedridden patients who can only receive AP radiographs and tend to be in worse condition with a higher rate of relevant diseases. Also, the study population of 477 included patients is relatively small when considering AI evaluation. Nevertheless, the study population offered a comprehensive overview of a middle-European population. Compared to other studies evaluating AI applications, the extensive gold standard reporting is a strong feature of the presented analysis. Most AI studies rely on pre-existing data from finished reports, which are more likely prone to containing wrong diagnoses, especially when data mining is performed automatically using only specific keywords, as is common. In this study, we implemented a two-step process of independent first readings with clearly structured reporting forms followed by a consensus reading for cases with initial discrepancies. The importance of this step can be seen in the relatively large number (203, 42.56%) of patients needing re-evaluation. Even though the two-step approach applied in this study is more extensive and allows less room for errors, the robustness of the “ground truth” is still the major flaw of this study, as is the case for the vast majority of AI studies in the field of medicine.

In an ideal research scenario, to provide a general and reliable benchmark, all AI applications and their results would first be tested against the same reference standard reports that have been confirmed by additional multiple and independent robust data, such as clinical examination results, laboratory results, pathology reports, and even autopsy reports for lethal outcomes, before undergoing in-house testing in each hospital prior to their deployment. However, for most studies, including this one, no unimpeachable gold standard is available, and testing multiple applications on the same dataset is beyond the capacities of single institutions. This method of evaluation, though for now without alternative, limits room for technological advancement and creates a necessity of careful interpretation of the terms “true” and “false”, when describing positive and negative findings: AI applications might actually be able to report diseases that radiologists might miss or, on the other hand, might be able to rule out diseases that radiologists might falsely classify as positive. These cases, however, are systematically discarded when evaluating AI against radiologists as a reference standard. This needs to be considered, because sensitivities and specificities of 100%, implying perfect image interpretation, then only imply that an AI performs exactly as good (or bad) as radiologists.

Since not all ML applications provide the same kind of probability values for their findings, the results from this study are not simply and completely transferable to all other ML applications. However, as most AI applications provide some kind of lesion probability in their reports, the general idea of this study will apply to the majority of ML systems currently on the market: Radiologists should not simply implement the factory settings of clinical AI applications, but they should consider adjusting them for specific tasks.

## 5. Conclusions

Even though CT diagnostics are becoming increasingly dominant in modern chest imaging due to their higher sensitivity in complex diseases, chest radiography remains one of the most commonly used imaging modality, and this is most likely to continue being the case. Despite its very low radiation exposure, little examination effort, and low cost, it renders crucial information on the vital organs of the heart and lungs and therefore offers a comprehensive overview of a patient’s condition, thus often being used as a front-line measure in cases of the deterioration of a patient’s status for unknown reasons.

The evaluated DL application showed good to excellent concordance with radiologists’ findings when applied for recognition of one single suspected diagnosis. It might enable a significant workload reduction of about 40% in ruling out specific diseases, thereby maintaining sensitivities above 95% and NPVs around 99% when reasonable optimized cutoff values for DL results are applied. For combined screening for multiple diseases, as would be necessary for automated comprehensive interpretation of chest radiographs, the concordance in the presented study dropped significantly. Therefore and as of today, DL applications appear not to be suitable for completely unsupervised application in the clinical setting. Presumably, this will continue to be the case for some time, even when keeping aside the general ethical and legal implications of autonomous AI-reporting.

In addition to general obstacles, bias against certain patient groups will have to be researched in further depth, and software developers need to implement effective solutions to counter discriminative tendencies in AI algorithms. To improve the overall quality of medical and especially radiological AI software, changes in policy and action in the scientific community should be pushed to establish and evolve large, international datasets with multimodally confirmed diagnoses and annotated image material for the training and testing of AI software.

The most feasible utilization of DL applications in the clinical setting at the moment appears not to be in providing precise reports for specific lesions but in providing reliable true negative reports for patients without findings. This could potentially reduce radiologists’ workload significantly and redirect sparse resources to those patients actually requiring doctors’ attention. Despite their current limitations, DL applications such as those examined in the present study can already be highly supportive in clinical routine and are likely to improve the accuracy of radiology reports. To do so, they need to be applied in the right setting and with the right threshold values, especially when different applications are combined. The utilization of AI, however, requires well-informed, critical supervising radiologists who are able to adapt AI to fit their specific needs (in thoracic imaging).

## Figures and Tables

**Figure 1 healthcare-12-00706-f001:**
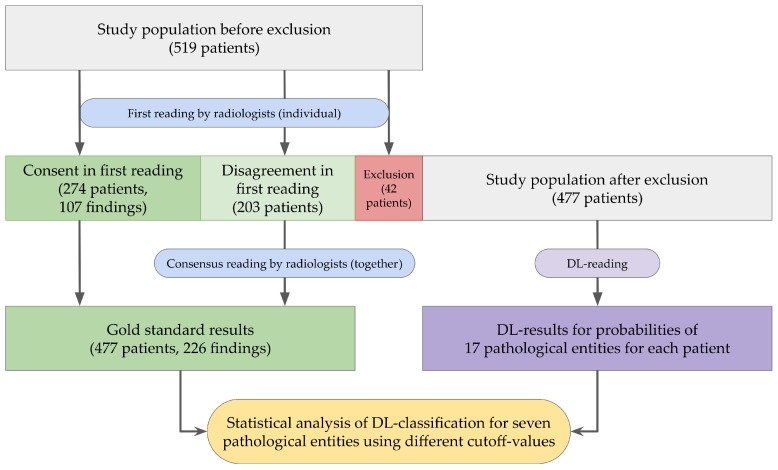
Flowchart of data processing for this study—A total of 519 consecutively performed chest radiographs in 2 planes were acquired. A total of 42 patients were excluded. The remaining 477 examinations were evaluated by two experienced radiologists (>5 years and >20 years of experience) for the gold standard. Additionally, InferRead DR Chest interpreted the images for 17 pathological conditions, of which the seven most clinically relevant were then compared with the gold standard findings for different classification cutoff thresholds.

**Figure 2 healthcare-12-00706-f002:**
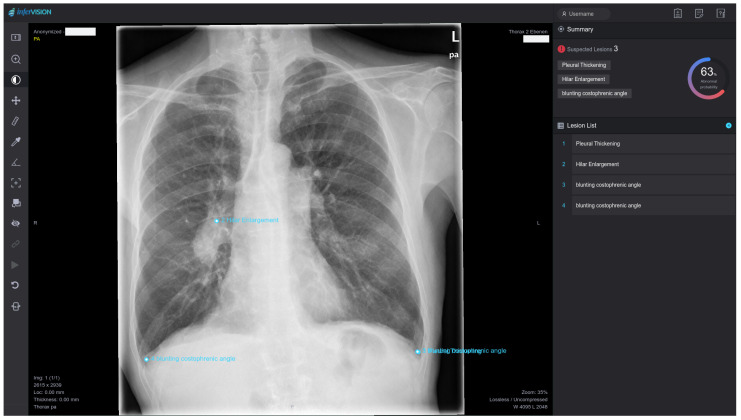
Example image of InferRead DR Chest Interface: Lesions classified as pathological by the software are topographically accurately marked blue. In the upper right part of the image, the application indicates the probability of any abnormality being present in the image. The markings can be suppressed, and overlapping labels can be moved by the user. In the lower right part of the image, a cumulative English report comprising all AI findings is provided, which can be edited. The image shows the findings for the standard factory setting without adjustments in threshold values.

**Figure 3 healthcare-12-00706-f003:**
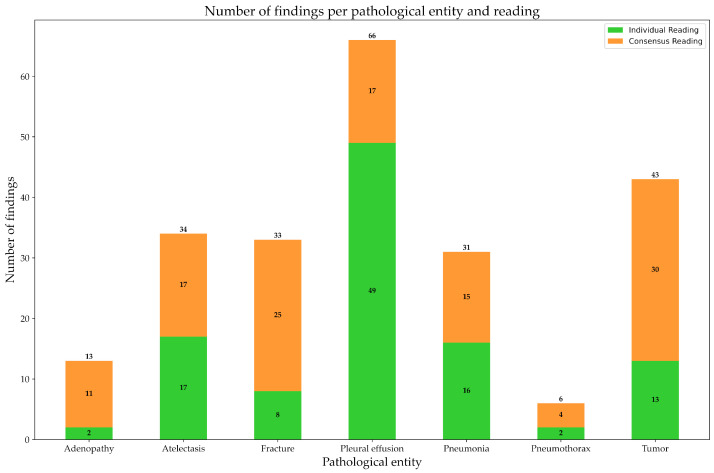
Visualization of all positive findings for each of the seven analyzed abnormalities divided by individual reading (green) and consensus reading (orange).

**Figure 4 healthcare-12-00706-f004:**
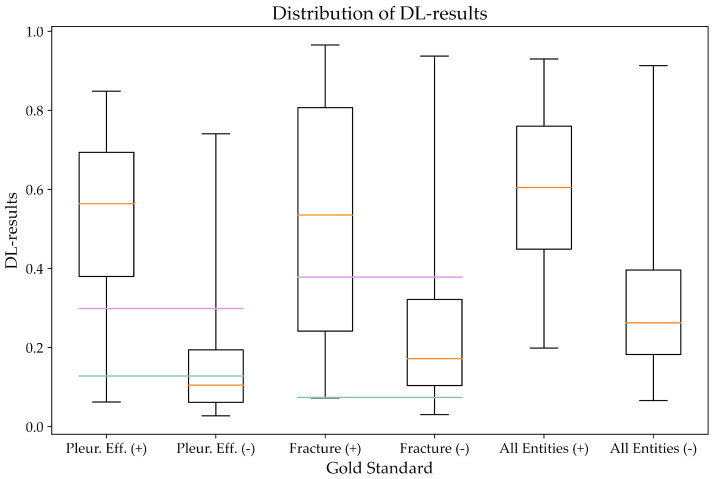
Distribution of DL results in gold standard positive and negative cases for pleural effusion and fracture, as well as the average across all evaluated abnormalities. The plot displays the minimum and maximum values (whiskers), the Q1 and Q3 (boundaries of the boxes), and the median (orange lines). The horizontal lines display cutoff values for classification into positive (above) and negative (below) findings, with the purple lines representing threshold values calculated applying optimization by Youden’s J and the green lines representing cutoff values for a sensitivity of 95% or higher. For gold standard positive cases, all patients with DL values above the cutoff line are classified true positive and those below it false negative. For gold standard negative cases, all patients below the cutoff line are classified true negative and those above it false positive.

**Figure 5 healthcare-12-00706-f005:**
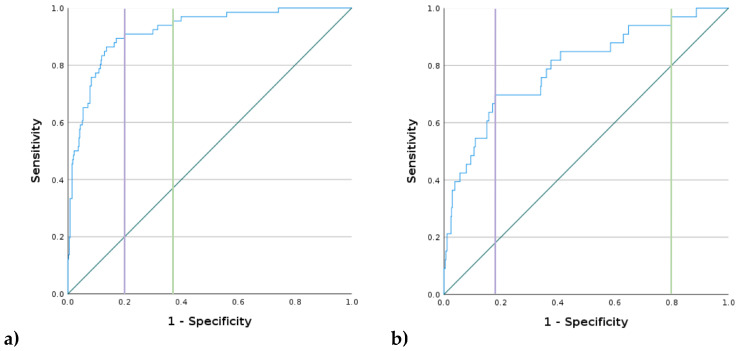
ROC curves for (**a**) pleural effusion and (**b**) fracture, with purple lines representing cutoff for optimization by Youden’s J and green lines representing sensitivity of 95% or higher.

**Figure 6 healthcare-12-00706-f006:**
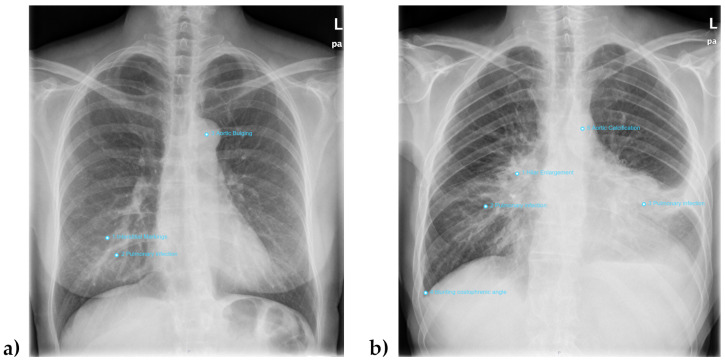
Example images of misclassified cases: Image (**a**) shows a 54-year-old female patient with no pathological findings in the gold standard analysis. The inferior bronchi on both sides are prominently featured but still within the normal range. The DL computed a probability value of 0.954652 for pneumonia in the right inferior part of the image. The other labels show DL findings for abnormalities not analyzed in this study (interstitial markings (false positive) and aortic bulging (true positive)). Image (**b**) shows the complex case of a 52-year-old male patient. While the DL correctly calculated high probabilities for adenopathy (0.880957) and atelectasis (0.716307), the DL value for tumor was just 0.390357. Instead, the application gave a falsely high value of 0.963765 for pneumonia, which, however, can be considered to be a differential diagnosis for the shade in the area of the left inferior lobe. The other labels again show DL findings for abnormalities not analyzed in this study (blunting of the right costophrenic angle (false positive) and aortic calcification (true positive)). Even though in this case the DL value for tumor was relatively low, the high value for adenopathy would have resulted in double-checking by a radiologist anyways, even for less conservative cutoff values.

**Figure 7 healthcare-12-00706-f007:**
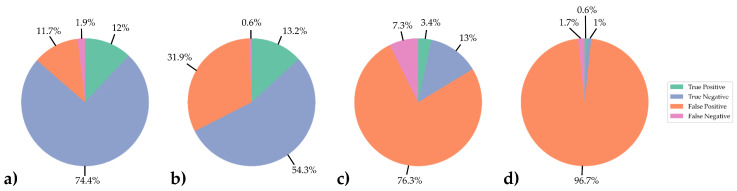
Percentages for classification of DL results for (**a**) the cutoff value (0.3083) applying optimization by Youden’s J and (**b**) the cutoff value (0.1429) for a sensitivity above 95% when screening for only pleural effusion, for which the DL application had achieved the highest AUC. Chart (**c**) shows the results for combined comprehensive reading for all seven pathological entities for every patient for cutoff values optimized by Youden’s J and (**d**) for cutoff values for sensitivities of 95% or higher.

**Table 1 healthcare-12-00706-t001:** Reasons for exclusion from final study population with absolute and relative frequency—insufficient lateral projections were by far the most common cause for exclusion.

Reason for Exclusion	Number of Cases (%)
Lateral image incomplete	33 (78.6)
Did not elevate arms for lateral image	2 (4.8)
Age (<18 years) ^1^	1 (2.4)
PA image incomplete	1 (2.4)
PA image missing	1 (2.4)
Did not elevate arms at all	1 (2.4)
Image does not match accession-ID	1 (2.4)
Insufficient inspiration	1 (2.4)
Lateral image is oblique	1 (2.4)

^1^ One tall 17 year old patient was examined using the “adult” setting on the X-ray machine. Therefore, the case was automatically included at first when the modality filter “chest radiograph 2 planes, adult” was applied for image acquisition.

**Table 2 healthcare-12-00706-t002:** Architecture of the U-Net convolutional neural network, a special type of machine learning programming for biomedical image segmentation, used in the InferRead DR Chest application.

Layer Name	Kernel	Number of Layers
Conv 1	3 × 3 64 stride 1	2
Conv 2	3 × 3 128 stride 2 3 × 3 128 stride 1	1 1
Conv 3	3 × 3 256 stride 2 3 × 3 256 stride 1	1 1
Conv 4	3 × 3 256 stride 2 3 × 3 256 stride 1	1 1
Conv 5	3 × 3 256 stride 2 3 × 3 256 stride 1	1 1

**Table 3 healthcare-12-00706-t003:** General characteristics of the study population regarding final population, as well as excluded patients.

	Male (%)	Female (%)	Age Range	Age Mean (SD)	Age Median (IQR)
**Final population** ^1^	284 (59.54)	193 (40.46)	18–91	61.4 (16.73)	63 (51–75)
**Excluded patients** ^2^	31 (73.81)	11 (26.19)	17–87	58.4 (15.87)	60.5 (50–69.5)

^1^ A total of 477 patients remained after application of exclusion criteria. ^2^ A total of 42 patients were excluded.

**Table 4 healthcare-12-00706-t004:** Absolute and relative numbers of all abnormalities found in the final gold standard and in the first and consensus reading.

Abnormality	Number of Cases (%) Final Gold Standard	Number of Cases (%) First Reading	Number of Cases (%) Consensus Reading
Adenopathy	13 (5.75)	2 (1.87)	11 (9.24)
Atelectasis	34 (15.04)	17 (15.89)	17 (14.29)
Fracture	33 (14.60)	8 (7.48)	25 (21.01)
Pleural effusion	66 (29.20)	49 (45.79)	17 (14.29)
Pneumonia	31 (13.72)	16 (14.95)	15 (12.61)
Pneumothorax	6 (2.65)	2 (1.87)	4 (3.36)
Tumor	43 (19.03)	13 (12.15)	30 (25.21)
All abnormalities	226	107	119
Number of patients	477	274	203

**Table 5 healthcare-12-00706-t005:** DL results for all cases with positive findings in the final gold standard reading for each pathological entity, as well as average result across all entities.

Abnormality	Median (IQR)
Adenopathy	0.6537 (0.5584–0.7211)
Atelectasis	0.5065 (0.428–0.6619)
Fracture	0.5353 (0.2416–0.8071)
Pleural effusion	0.5639 (0.3798–0.6938)
Pneumonia	0.6953 (0.5705–0.8207)
Pneumothorax	0.657 (0.5016–0.8133)
Tumor	0.621 (0.4624–0.8029)
Average across all entities	0.6047 (0.4489–0.7601)

**Table 6 healthcare-12-00706-t006:** DL results for all cases with negative findings in the final gold standard reading for each pathological entity, as well as average result across all entities.

Abnormality	Mean (SD)	Median (IQR)
Adenopathy	0.4668 (0.3357–0.5979)	0.4556 (0.3696–0.5599)
Atelectasis	0.2358 (0.0500–0.4216)	0.1738 (0.1067–0.2964)
Fracture	0.2417 (0.0465–0.4369)	0.1718 (0.1038–0.3216)
Pleural effusion	0.1581 (0.0170–0.2992)	0.1048 (0.0612–0.1940)
Pneumonia	0.3747 (0.1474–0.6020)	0.3403 (0.1924–0.5368)
Pneumothorax	0.2695 (0.0923–0.4467)	0.209 (0.148–0.3325)
Tumor	0.4230 (0.2510–0.5950)	0.3826 (0.2945–0.5318)
Average across all entities	0.3099 (0.1342–0.4856)	0.2625 (0.1823–0.3962)

**Table 7 healthcare-12-00706-t007:** DL thresholds and their resulting classification outcomes when optimization by Youden’s J was applied for cutoff calculation. Results are presented for each pathological entity, as well as for the average across all entities.

Metric	Adenopathy	Atelectasis	Fracture	Pleural Eff.	Pneumonia	Pneumothorax	Tumor	Average
Cutoff value	0.4891	0.3375	0.3853	0.3083	0.5215	0.4876	0.4173	0.421
AUC (95%-CI)	0.845 (0.76–0.93)	0.866 (0.82–0.92)	0.793 (0.71–0.88)	0.92 (0.89–0.96)	0.839 (0.78–0.90)	0.843 (0.65–1.00)	0.791 (0.73–0.86)	0.842 (0.76–0.92)
Sensitivity	100%	82.4%	69.7%	86.4%	87.1%	83.3%	88.4%	85.3%
Specificity	58.0%	80.6%	82.0%	86.4%	73.5%	88.1%	57.1%	75.1%
TP (%)	13 (2.73)	28 (5.87)	23 (4.87)	57 (11.95)	27 (5.66)	5 (1.05)	38 (7.97)	27.29 (5.72)
TN (%)	269 (56.39)	357 (74.84)	364 (76.31)	355 (74.42)	328 (68.76)	415 (87.00)	248 (51.99)	333.71 (69.96)
FP (%)	195 (40.88)	86 (18.03)	80 (16.77)	56 (11.74)	118 (24.74)	56 (11.74)	186 (38.99)	111 (23.27)
FN (%)	0 (0)	6 (1.26)	10 (2.10)	9 (1.89)	4 (0.84)	1 (0.21)	5 (1.05)	5 (1.05)
PPV	6.25%	24.56%	22.33%	50.44%	18.62%	8.20%	16.96%	21.05%
NPV	100%	98.35%	97.33%	97.53%	98.80%	99.76%	98.02%	98.54%
FPR	42.03%	19.41%	18.02%	13.63%	26.46%	11.89%	42.86%	24.96%

**Table 8 healthcare-12-00706-t008:** Detailed classification results for combined comprehensive readings of all seven analyzed pathological entities in every chest radiograph. Results are given for threshold values optimized by Youden’s J and for threshold values for sensitivities above 95% per pathological entity in the exploratory analysis.

Metric	Optimization by Youden’s J	Exploratory Analysis (>95%)
Sensitivity	31.37%	27.27%
Specificity	14.55%	1.07%
TP (%)	16 (3.35)	3 (0.63)
TN (%)	62 (13)	5 (1.05)
FP (%)	364 (76.31)	461 (96.65)
FN (%)	35 (7.34)	8 (1.68)
PPV	4.21%	0.65%
NPV	63.92%	38.46%
FPR	85.45%	98.93%

**Table 9 healthcare-12-00706-t009:** DL thresholds and their resulting classification outcomes from applying sensitivities above 95% for each abnormal entity for cutoff calculation in the exploratory analysis. Results are presented for each pathological entity, as well as the average across all entities.

Metric	Adenopathy	Atelectasis	Fracture	Pleural Eff.	Pneumonia	Pneumothorax	Tumor	Average
Cutoff value	0.4891	0.1978	0.0902	0.1429	0.2976	0.1602	0.3412	0.2456
Sensitivity	100%	97.1%	97.0%	95.5%	96.8%	100%	95.3%	97.4%
Specificity	58.0%	55.3%	20.0%	63.0%	44.2%	30.1%	40.6%	44.5%
TP (%)	13 (2.73)	33 (6.92)	32 (6.71)	63 (13.21)	30 (6.29)	6 (1.26)	41 (8.6)	31.14 (6.53)
TN (%)	269 (56.39)	245 (51.36)	89 (18.66)	259 (54.3)	197 (41.3)	142 (29.77)	176 (36.9)	196.71 (41.24)
FP (%)	195 (40.88)	198 (41.51)	355 (74.42)	152 (31.87)	249 (52.20)	329 (68.97)	258 (54.09)	248 (51.99)
FN (%)	0 (0)	1 (0.21)	1 (0.21)	3 (0.63)	1 (0.21)	0 (0)	2 (0.42)	1.14 (0.24)
PPV	6.25%	14.29%	8.27%	29.30%	10.75%	1.79%	13.71%	11.16%
NPV	100%	99.59%	98.89%	98.85%	99.49%	100%	98.88%	99.42%
FPR	42.03%	44.70%	79.95%	36.98%	55.83%	69.85%	59.45%	55.77%

## Data Availability

Data are contained within the article.

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
