# Peer review of "Doctor’s Orders—Why Radiologists Should Consider Adjusting Commercial Machine Learning Applications in Chest Radiography to Fit Their Specific Needs"

_healthcare, 2024, doi:10.3390/healthcare12070706_

Round 1

Reviewer 1 Report

Comments and Suggestions for Authors

Deep learning has found its popularity in NLP and computer vision, with prosperous applications in medical diagnosis. This paper focused on the Chest Radiographs, and by comparison the deep learning based commercial software with expert, this paper comes to a conclusion that Deep learning-based AI can be employed for clinical applications. This comparison study is worthy to be published to advance the development of AI. The authors also reported some biases, my concerns are:

1)       The amount and distribution of the training dataset are crucial to the performance of the software, the biases may be resulted by the dataset. The authors may say something about the commercial software.

2)       It is better if some cases that the AI failed to diagnosis are presented and some discussion is launched.

Author Response

Dear reviewer,

thank you very much for your constructive suggestions to improve our paper!

We did our best to include your ideas into the updated version of the manuscript.

More specifically:

  1. We contacted the manufacturer once more to get additional information on the architecture and training data. Due to commercial reasons they naturally could not give us the code of their software but we feel like were able to add extensive information on the applications architecture and training data, which we hope will provide a substantial addition to the paper’s material and methods section.
  2. As per your suggestion, we also added another image panel, which includes 2 cases in which the AI provided (either) false positive and/or even false negative findings.

We hope that you find the updated manuscript improved to your satisfaction.

Best regards

Reviewer 2 Report

Comments and Suggestions for Authors

Dear Authors,

Your manuscript titled "Doctor’s Orders – Why Radiologists Need To Adapt AI To Fit Their Specific Needs" is exceptionally well-written, clear, and engaging. The statistical analyses are beautifully reported, offering a thorough evaluation of commercial tools designed to automatically detect chest radiographs.

While there are no major issues, the manuscript could be further enhanced. Please find a minor comment:

In the introduction section, lines 42-46 mention "a lot of attention by researchers and companies in radiology towards developing solutions, especially focusing on engineering, mainly Deep Learning (DL) based, artificial intelligence (AI) models for automated image classification." It is recommended that the authors also mention recent trends in AI/ML applications in radiology devices that are already available. You might consider citing this article [https://www.medrxiv.org/content/10.1101/2022.12.07.22283216v3.full-text], which provides insights into AI/ML utilization across various medical sub-specialties. This reference also highlights radiology as a leading subspecialty where AI has made significant strides in recent years. This will further enrich the discussion section.

I hope this suggestion will aid in strengthening your manuscript.

Best regards.

Author Response

Dear reviewer,

thank you very much for your constructive suggestions to improve our paper!
We did our best to include your ideas into the updated version of the manuscript.

More specifically:

  • As per your suggestion we included information from the paper you provided and added a short paragraph describing recent trends in radiology in lines 49 to 53.
  • We also included one of the statistics on third-party-testing from the paper at the end of the introduction in lines 70 and 71.

We hope that you find the updated manuscript improved to your satisfaction.

Best regards

Reviewer 3 Report

Comments and Suggestions for Authors

In this study, the authors investigated the performance of Deep Learning (DL) software in different possible application scenarios for automatic comprehensive interpretation of chest radiographs. This study has been well handled by the authors and all the details of the study have been carefully composed. However, I have only a few suggestions for this study:

1. The abstract part of the study is written too long. This section can be shortened.

2. Authors should explain why the DL model/algorithm is preferred as a method.

3. Before using the DL algorithm, descriptive statistics data of the dependent and independent variables in the data set should be shared (between Line 186-187: Table 2 is insufficient).

4. Correlation data of the variables should be obtained.

5. If possible, ANOVA data should be obtained and the effect of independent variables on the dependent variable should be examined (the p-value of the variables should be calculated).

6. The difference between the two algorithms can be expressed using another artificial intelligence (or machine learning) algorithm.

Author Response

Dear reviewer,

thank you very much for your constructive suggestions to improve our paper!

We did our best to include your ideas into the updated version of the manuscript.

More specifically:

  1. We shortened the abstract part and condensed its information to only contain the very most relevant parts of the study.
  2. We added a paragraph about the benefits of DL algorithms as opposed to conventional “rule-based” programming in the materials and methods section in lines 149 to 152 of the updated manuscript.
  3. We added descriptive statistics on the deviation of the variables age and sex in the gold standard positive and negative cases in lines 226 to 231 to provide more details and a better understanding of the study’s data set.
  4. We considered performing a correlation analysis in the study design but decided against it for two reasons:
    1. Before performing the sequential regression for analyzing the influence of the factors age, sex, external material and multimorbidity, we tested our data using the variance-inflation-factor to rule out collinearity. Results were between 1.041 and 1.068 for all variables, which indicates that there was no relevant collinearity (as is assumed for results below 10.00). Correlation analysis in this case would not have generated significant additional information.
    2. We evaluated the level of concordance between gold standard findings and AI-results using ROC-analysis, which provides more information than correlation analysis would.
  5. Due to the different scale levels of the variables ANOVA could not be applied in this study. Instead we used the sequential regression model for analyzing the influence of the independent variables on the dependent variable, which is DL-result. We provided the p-values for the model in the results-part of the manuscript in lines 333 to 338 in the updated manuscript.
  6. Only one machine learning model/software was applied in this study. The two separate segments in the results-section derive from using of two different thresholds for cutoff when classifying AI-results into “positive” and “negative”. The more “conservative” threshold required sensitivities of 95% or higher, resulting in a lower specificity and creating more false positive cases. The other one used optimization by applying Youden’s J for an optimized balance between sensitivity and specificity.

We hope that you find the updated manuscript improved to your satisfaction.

Best regards

Reviewer 4 Report

Comments and Suggestions for Authors

The followings are the major concerns for this study: 

- The title of manuscript needs to be revised to specifically depict the focus of this study and the experiments conducted. (specific modal used; specific groups of symptoms/diseases; specific DL)

- The so called 'DL' in this study needs to be specifically referred. 

- Actually, the merit of this study is limited. Because there are several CNN/DL models developed and applied for biomedical images, various CNN or DL models needs to be considered and applied. 

Author Response

Dear reviewer,

thank you very much for your suggestions to improve our paper.

We did our best to include your ideas into the updated version of the manuscript.

More specifically:

  • We added the modality of imaging to the title and specified the message from the initial broad statement to a more narrow and focused one.
  • We contacted the software’s manufacturer once more and were now able to provide additional information on the architecture and training data of the DL-model in the material and methods section, especially in lines 149 to 173 and table 2 of the updated manuscript.
  • Concerning your last point:
    • we added a paragraph explaining that not all of the findings are universally transferable to other Machine Learning models in lines 525 to 530. However, since the very most of Machine Learning applications currently on the market will provide some kind of probability value to enable radiologists to interpret the findings rather than just relying on the blackboxed results, we do feel that the general idea of sensitizing radiologists to adjusting ML-thresholds depending on their field of application is relatively widely transferable. We did phrase this part out more in the updated version of the manuscript.
    • You suggested testing our dataset on multiple available applications, which we agree would be ideal for comparison. Unfortunately this is beyond the scope and capacities of our study. However, this dilemma is why in the discussion part we advocate for international, universally available and validated datasets to test (at best) all available applications on as a benchmark. But even then as a measure of best practice each hospital should be performing additional in-house-testing on the application of their clinical choice. We tried to phrase this part out more as well in the revised version, especially in lines 507 to 516.

We hope that you find the updated manuscript improved to your satisfaction.

Best regards

Reviewer 5 Report

Comments and Suggestions for Authors

1. The authors have to clearly write the contributions point-by-point in the at the end of introduction section.

2. Need additional design details using DL

3. In case if you used multiple models, give details

4.DL analysis: need more details. Include architectural details 

5. A separate label is missing; all the figure label are expected to be provided separately from the paragraph

6. Provide interpretations on figure 4

7. The discussion in the result must have comparison study of proposed method with related work

8.Discuss  limitations and future scope

9. The tile is also too broad; it is recommended to modify with more specific details

Comments on the Quality of English Language

Suggested for minor check 

Author Response

Dear reviewer,

thank you very much for your constructive suggestions to improve our paper!

We did our best to include your ideas into the updated version of the manuscript.

More specifically:

  1. We tried to better phrase out the goals of this study in the introduction section, especially in lines 72 to 81 of the updated manuscript.
  2. We contacted the manufacturer once more and were able to provide additional information on the application’s architecture and training data for the material and methods section, mainly in lines 149 to 173 and table 2 of the updated manuscript.
  3. We did not use different models. Only one machine learning model/software was applied in this study. The two separate segments in the results-section derive from using of two different thresholds for cutoff when classifying AI-results into “positive” and “negative”. The more “conservative” threshold required sensitivities of 95% or higher, resulting in a lower specificity and creating more false positive cases. The other one used optimization by applying Youden’s J for an optimized balance between sensitivity and specificity.
  4. See point 2.
  5. We tried to improve the labels for our figures and provide more information there.
  6. We tried to give a more detailed interpretation of figure 4 in lines 242 to 246 of the updated manuscript and edited parts of the figure’s description, trying to point out especially the relevance of the colored cutoff-values, as their role is crucial for the paper’s message and is the main part of the results section and the basis of the paper’s discussion segment.
  7. We tried to add more detailed information on the results of similar applications when screening for one single abnormality in lines 351 to 355. While of course today there are many applications for the detection of single specific diseases, to the best of our knowledge as of now the scope of software solutions for comprehensive readings of chest radiographs for sorting out “normal” cases is limited to the applications evaluated by Yoon et al., Dyer et al. and Plesner et al., which we compared our findings to in lines 407 to 431.
  8. We added a paragraph explaining that not all of the findings are universally transferable to other Machine Learning models in lines 525 to 530. We tried to describe obstacles and opportunities for ML-applications in the clinical setting in lines 545 to 563.
  9. We edited the title and formed a more narrow and specific statement instead of the initial broad one.

We hope that you find the updated manuscript improved to your satisfaction.

Best regards

Round 2

Reviewer 4 Report

Comments and Suggestions for Authors

The revised manuscript is in an acceptable form.